# Can participatory budgeting mitigate government debt risk？—An empirical analysis using cross-national panel data

**Yongpeng Li**[1]*, **Qianqian Zhang**[2]

**1** School of Business, Huaiyin Normal University, Huai'an, Jiangsu Province, China, **2** School of Business, Beijing Technology and Business University, Beijing, China

* 1507308616@qq.com

## Abstract

### Objectives

Participatory budgeting, serving as a complementary mechanism to traditional democratic practices, grants citizens direct involvement in the budgetary decision-making process, thereby embodying features of direct democracy. It has consequently gained recognition and promotion in numerous countries. However, scholarly opinion remains divided regarding the efficacy of participatory budgeting in mitigating government debt risks. This study seeks to address this ongoing academic debate by employing a cross-national panel data.

### Methods

Utilizing a dataset of 664 samples from 83 countries between 2008 and 2023, sourced from the International Budget Partnership (IBP), this study combines survey data on public participatory budgeting opportunities with government debt statistics from the International Monetary Fund (IMF). Fixed-effects and mediating effect models are employed to empirically examine the relationship between public budget participation and government debt risk.

### Results

Our analysis indicates that, on average, government debt risk is markedly lower in cases of participatory budgeting, a finding that is significant at the 1% level. This finding remains robust after replacing the explained variable to address robustness concerns and employing instrumental variables to mitigate endogeneity issues. Mediation analysis further indicates that the quality of explicit debt disclosure serves a significant mediating role, whereas contingent liabilities do not exhibit a mediating effect. Additionally, the mitigating effect of participatory budgeting on government debt risk

**Data availability statement:** All the data in the research can be found at the following links. 1.Government debt risk data: https://www.imf.org/external/datamapper/datasets/FM 2.Budget participation data: https://internationalbudget.org/open-budget-survey/download 3.Controlled variable data: https://datatopics.worldbank.org/world-development-indicators/.

**Funding:** The authors received specific funding for this work from the Chinese Social Science Foundation Project (Grant No.: 24BJY204; Principal Investigator: Yongpeng Li ).

**Competing interests:** The authors have declared that no competing interests exist.

is more pronounced in countries adopting accrual-based government accounting compared to those using cash-based accounting.

## 1 Introduction

Recently, rising sovereign debt levels across Asia, Europe, South America, and Africa have led to widespread credit rating downgrades, exacerbating fiscal vulnerability. According to the International Monetary Fund (IMF), global government debt will reach a historic $100 trillion by the end of 2024, equivalent to 93% of the world's GDP—a 10 percentage point increase from pre-pandemic levels in 2019. This trend, highlighted in the IMF working paper "Global Public Debt: Probably Worse Than It Looks" [1], underscores the urgent need for coordinated policy responses. Consequently, developing effective mechanisms to mitigate sovereign debt risk has emerged as a critical challenge that demands immediate international attention.

Since the advent of the New Public Management movement, public engagement has evolved from a primary emphasis on service quality evaluation to active participation in governance through formal avenues such as public finance consultation and organized interest representation [2]. Participatory budgeting (PB) stands as a prime manifestation of this shift. Originating in Porto Alegre, Brazil, PB serves as a complement to traditional elite democracy, as it enables citizens to directly engage in the allocation of budgetary resources—reflecting distinct characteristics of direct democracy [3]. In this context, globally proliferating PB has emerged as a viable institutional vehicle that has been demonstrated to enhance budgetary transparency [4,5], improve fiscal efficiency [6], and reallocate spending toward public preferences [7]. However, whether it mitigates government debt risk remains debated. Some scholars, drawing on Keynesian theory, argue that participants (especially those from lower-income backgrounds) often operate under a "fiscal illusion," believing that expansion of government debt could increase their current disposable income, and thus tend to support debt accumulation [8].For instance, a study of South Korean local governments (2012–2019) found that greater public engagement in PB correlated with worse local fiscal health [9]. On the other hand, scholars grounded in rational expectations theory contend that rational participants are able to perceive the future burden of debt repayment and would not be misled by such an illusion [10]; as a result, they would oppose excessive government borrowing. For instance, a study of 155 Slovak municipalities found a weak inverse relationship between PB and fiscal risk [11]. In Wenling, China, the practice of PB has shown that by subjecting fiscal decisions to public scrutiny, it creates an effective debt restraint, significantly improving the financial condition of pilot townships [12]. Specifically, PB provides an effective pathway for improving local government fiscal conditions by enhancing fiscal transparency and cultivating public budget literacy [13]. This leads to the central question of whether PB can mitigate government debt risk, and if so, through which channels; if not, what factors explain its ineffectiveness. This study aims to empirically examine this ongoing debate.

It should be noted that the existing literature typically defines PB as a mechanism for direct citizen engagement in budget decisions at local government levels (such as cities or counties), focusing on resource allocation for specific jurisdictions in areas like community projects or infrastructure. In contrast, this study examines PB as defined by the International Budget Partnership's Open Budget Survey (OBS), where it serves as an indicator of national-level budget transparency. Through mechanisms like public suggestions, expert consultation, and feedback, PB helps reveal potential fiscal risks, improve the evidence-based nature of the budget, and foster public debate on long-term pressures such as pensions and healthcare—thereby strengthening societal oversight of fiscal sustainability. Table 1 presents a comparative analysis of local and national participatory budgeting, focusing on their differential effects on government debt risk.

As shown in Table 1, although the two forms differ in scope and process, they share fundamental goals: democratizing budget decisions, strengthening public influence over fiscal resources, and advancing more efficient and equitable public resource management.

This study aims to investigate whether and how citizen participation in public budgeting affects a country's government debt risk. For this purpose, we employ the OBS score for citizen budget participation opportunities as a proxy for participatory budgeting (PB). The theoretical framework does not strictly distinguish between local and national-level participation; rather, it conceptualizes citizen budget participation comprehensively to analyze its macro-level relationship with government debt risk.

This study makes dual contributions:

(1) Pioneering empirical evidence demonstrating the efficacy of PB in containing fiscal risks and overcoming prior studies' reliance on qualitative approaches.

(2) Reveal systematic underreporting of contingent liabilities by integrating debt transparency into a PB risk-mitigation framework.

The remainder of this paper is structured as follows. Section 2 outlines the research hypotheses. Section 3 describes the research design and data. Section 4 presents the empirical results. Section 5 investigates the mechanism through which PB mitigates government debt risk. Section 6 and 7 discuss the findings and present policy recommendations. Finally, Section 8 concludes with the study's limitations and directions for future research.

## 2 Research hypotheses

Originating in Porto Alegre, Brazil, PB embodies deliberative democracy by enabling direct citizen engagement in local fiscal processes, a departure from bureaucratic budget monopolies [14,15]. The PB moves beyond conventional citizen roles as voters or watchdogs, transforming participants into co-producers of budgetary policy [16,17]. Its social accountability

**Table 1. Local versus National Participatory Budgeting.**

| Comparison Dimension | Local-Level Participatory Budgeting | National-Level Participatory Budgeting |
|---|---|---|
| Core Definition | It is defined as the public's participating in the entire budgeting process of grassroots governments, from formulation to oversight, thus promoting democratic and deliberative management. | It is defined as a formal opportunity for public participation in the budget process, which is a key pillar of fiscal accountability. |
| Typical Scope of Application | It is primarily used by grassroots governments to fund specific local projects—such as pocket parks and sewage treatment. | It oversees the entire budget process, focusing on high-level fiscal planning and priority-setting for major departmental budgets. |
| Impact on Fiscal Risks | This process safeguards public resources through correcting unreasonable expenditures in the draft budget and rationalizing financial allocation priorities. | These mechanisms enhance budgetary rigor and stimulate public debate on long-term fiscal pressures (e.g., pensions, healthcare), thereby strengthening societal oversight of fiscal sustainability. |

mechanism demonstrates distinct effectiveness in reducing sociopolitical exclusion compared to vertical or horizontal accountability within closed political systems [18].

Budget participants evaluate government debt arrangements primarily through the lens of self-interest, focusing on consequences for their future income, employment opportunities, and other pecuniary and material gains.

Regarding the relationship between government debt and resident income, existing research remains divided on whether public debt can actually raise residents' income. Supporters argue that budget participants, often facing liquidity constraints, tend to be highly sensitive to current disposable income while underestimating the future tax burden associated with debt financing. As a result, they are generally inclined to support expansion of government debt. Panizza & Presbitero [19] suggested that an expansion of government debt could lead residents to perceive an increase in their current income, thereby stimulating consumption. Romer & Romer [20] also found that raising transfer payments—potentially funded through bond issuance—significantly improved recipients' income and consumption levels, particularly among low-income groups. Their findings support the positive role of expansionary fiscal policies, including debt financing, in enhancing equity and raising incomes for vulnerable populations. However, other scholars hold opposing views. Barro [21] contended that since the government does not generate wealth itself, public debt must ultimately be repaid through taxation. He argued that rational consumers, anticipating future tax hikes to repay debt, would save rather than spend any current tax cuts resulting from debt-financed policies, rendering the net effect of government debt on household income neutral. Peter [22] argued that government debt could crowd out capital accumulation, lower the capital-labor ratio in the steady state, and consequently reduce wage income. As a result, asset-holding elderly individuals might support debt expansion, whereas labor-dependent younger generations would likely oppose excessive government borrowing. Obiero & Topuz [23] find that both internal and public debt exacerbate long-term inequality in Kenya. Consequently, they recommend prioritizing non-debt financing options, as debt financing is deemed not pro-poor.

There is significant divergence in existing research regarding whether government debt affects residents' employment. Dawood et al. [24] argued that government debt expansion can stimulate the economy by boosting effective demand, thereby increasing short-term employment and garnering public support for such policies. Extending this view, Liu & Zeng [25] proposed that using debt funds for infrastructure projects—such as in transportation, agriculture, forestry, and water conservancy—can lower labor transfer costs, enhance agricultural productivity, release surplus labor, attract corporate investment, and create additional job opportunities, thereby effectively promoting non-agricultural employment. However, Bai et al. [26] questioned such perspectives, contending that government debt could produce a "crowding-out effect" that suppresses private investment, reduces labor demand, and ultimately adversely affects the employment rate. Shao et al. [27] similarly expressed caution, suggesting that the expansion of local government debt might attract highly skilled labor to concentrate in the public sector and financial industries, creating a "siphoning effect" that could hamper employment demand in the real economy.

There is significant divergence in the existing literature regarding whether government debt enhances residents' well-being. Min & Zhang [28] contend that government debt expansion can increase public expenditure, redirecting status-driven private consumption toward more inclusive and universally accessible public services, thereby contributing to higher overall citizen happiness. Park [29] further found that larger governments tend to exhibit greater expenditure transparency, which helps curb corruption and increases public satisfaction with government performance. However, opposing studies argue that rapid growth in government debt may promote the pursuit of prestige projects driven by performance metrics, resulting in inefficient use of fiscal resources and ultimately reducing residents' tangible sense of benefit [30]. Zhang & Kong [31] found that rising local government debt in China exacerbated urban environmental pollution in the short term, as substantial debt capital flowed into real estate and infrastructure construction. This boosted demand in high-energy consumption industries—such as steel and cement—while simultaneously crowding out environmental protection budgets. Liu et al. [32] propose a "dual nature" of debt: on the one hand, debt repayment pressure may crowd out government environmental expenditure, thereby weakening corporate incentives for environmental investment—a

mechanism referred to as the crowding-out effect; on the other hand, when debt funds are allocated to environmental projects, they may produce a guiding effect that stimulates follow-up investment from enterprises. The study concludes, however, that the crowding-out effect predominates. Similarly, Bökemeier & Greiner [33] suggest that when public debt finances non-productive expenditure, it may aggravate debt burdens and compromise future welfare spending. Table 2 provides a condensed overview of the major viewpoints in the scholarly debate regarding public budget participation and government debt.

Based on the above analysis, this study proposes the following competing hypotheses:

Hypothesis 1: To enhance income, employment, and sense of gain, budget participants tend to support the expansion of government debt, suggesting that the level of budget participation is positively correlated with government debt risk.

Hypothesis 2: Due to concerns over future tax increases and welfare reduction, budget participants tend to oppose the expansion of government debt, indicating that the level of budget participation is negatively correlated with government debt risk.

## 3 Research design

### 3.1 Variable definition

**3.1.1 Dependent variable.** Internationally, two standard indicators are used to assess government debt risk: namely, the debt-to-revenue ratio and the debt-to-GDP ratio. The debt-to-revenue ratio is calculated by dividing outstanding government debt by current government revenue, while the debt-to-GDP ratio is determined by dividing outstanding government debt by nominal GDP (Given that the IMF's statistical reporting framework defines general government data as the consolidation of central government, local government, and social security fund data, utilizing general government figures enables a more comprehensive assessment of sovereign debt risks. As evidenced by the IMF's evaluation of the Greek debt crisis—which mandated general government debt metrics inclusive of local governments' implicit liabilities rather than central government data alone—this approach captures systemic fiscal exposures. Accordingly, this study employs general government debt data as the proxy for sovereign debt risk exposure). This study uses the debt-to-GDP ratio as a proxy for government debt risk. The data for these indicators were sourced from the IMF's Government Finance Statistics (Data source: https://www.imf.org/external/datamapper/datasets/FM).

**3.1.2 Explanatory variables.** Since 2006, the International Budget Partnership (IBP) has conducted a biennial budget openness survey covering approximately 100 countries worldwide, with a three-year gap between 2012 and 2015. The most recent survey will be conducted in 2023. The Open Budget Survey (OBS) evaluates opportunities for public participation in the budgeting process on a scale of 0–100, where higher scores indicate greater public involvement, and lower scores reflect less participation. The survey specifically examines various dimensions of budget transparency and public engagement.

Table 2. Do Budget participants support or oppose government debt expansion.

| Dimension | Support Expansion | Oppose Expansion |
|---|---|---|
| Income level | Debt expansion may boost household income. | Rational individuals anticipate higher future tax burdens. |
| Employment | Debt expansion stimulates public and private investment, promoting job creation. | Debt expansion triggers a "siphoning effect" of high-skilled talent to the public sector, harming employment. |
| Sense of gain & Well-being | Debt-driven fiscal spending improves public infrastructure and enhances well-being. | Debt expansion often leads to vanity projects, environmental pollution, and fiscal waste, reducing residents' sense of benefit and happiness. |

For the 2006–2012 period, the survey included between the 114th and 125th questions related to the opportunity score, totaling 12 questions; for 2015, the survey included Question 114 and sequential Questions 119–133, totaling 16 questions; and for the 2017–2023 period, it ranged from the 125th to 142nd questions, totaling 18 questions. To ensure the accuracy of the public budget participation opportunity score, the OBS uses the questions related to budget participation to measure the extent to which governments involve the public in budget decision-making and monitoring. The responses for each thematic area are averaged, and each area receives a separate score. Additionally, the IBP collects information on the role of independent fiscal institutions (IFIs), which are nonpartisan bodies typically affiliated with the executive or legislative branch that produce fiscal forecasts and estimate policy costs.

The annual score was calculated by summing the scores for each question related to participation opportunities, and dividing by the total number of relevant questions. This score reflected the annual level of public participation in the budgeting process.

Due to the limited number of countries participating in the 2006 survey (only 59, increasing to 85 in 2008), this study used the 2008 as the starting year to maximize sample size. Excluding Sudan (which did not continue participating after 2012) and Afghanistan (for which data for explanatory and control variables were difficult to obtain), this study selected 664 sample observations from 83 countries spanning 2008–2023(Data source: https://internationalbudget.org/open-budget-survey/download).

**3.1.3 Set of control variables (X).** Active engagement of the legislature in the budget process is essential [34]. This represents people monitoring and reviewing the government's public debt. National government audit institutions are the final reviewers and guarantors of the smooth implementation and effectiveness of government budgets. They monitor and audit whether the use of government debt provides "value for money" [35].

Existing research also suggests that a country's government debt risk is closely linked to its economic growth [36]. Factors such as national income levels, urbanization rates, and the education levels of residents can also influence the scale and risk of government debt [37–39]. Based on these insights, the following variables were included as control variables in the model:

Economic growth rate, national income level (natural logarithm applied), residents' educational level data and urbanization rate data were taken from the World Bank (Data source: https://datatopics.worldbank.org/world-development-indicators/).

Legislative and audit supervision level data were obtained from the International Budget Partnership's OBS conducted between 2008 and 2023(Data source: https://internationalbudget.org/open-budget-survey/download).

**3.1.4 Robustness test and endogeneity test variables.**

(1)    Robustness test variables

Surrogate dependent variables. First, we replace the dependent variable with the debt-to-revenue ratio *(Deb_rev)*. As the IMF does not directly provide data for this ratio, it is computed using the following formula: Debt/Revenue Ratio = (Debt/GDP Ratio)/ (Revenue/GDP Ratio) × 100%.

Second, we use the debt-to-spending ratio (*Deb_spe*) as another alternative dependent variable, calculated as: Debt/Spending Ratio = (Debt/GDP Ratio)/ (Spending/GDP Ratio) × 100%. These fiscal data were derived from the IMF's October 2024 Global Fiscal Monitor report.

(2) Variables for Endogeneity Test. Following prior research, this study adopted the one-period lagged opportunity score for public budget participation as a proxy variable. Specifically, we used the opportunity score for public budget participation from 2006 to 2021. The missing data for 2006 were supplemented using linear interpolation.

## 3.2 Model setting

To investigate the relationship between public budget participation and government debt risk, this study draws on prior research [35] and construct the following benchmark measurement equation:

$$Deb_{it} = \theta_0 + \theta_1 part_{it} + \sum \theta_j X_{it} + \mu_i + \varepsilon_{it}$$

(1)

In the above system of equations, the subscript ($i$) represents the country and ($t$) represents the year. In Model (1), the variable ($Deb$) represents government debt risk, the variable ($part$) represents the level of public budget participation, and the variable ($X$) is a set of control variables that may influence government debt risk. These control variables include the country's economic growth rate ($Rgdp$), national income level ($LnGNI$), legislative supervision level ($Leg$), audit supervision level ($Aud$), urbanization rate ($Urb$), and residents' educational attainment ($Cul$). The terms $\mu$ are individual fixed effects, and $\varepsilon$ is the error term. Based on the aforementioned theoretical analysis, this study predicts that the coefficient ($\theta_1$) will be negative.

## 4 Empirical analysis

### 4.1 Descriptive statistical analysis

Descriptive statistics for the main variables are presented in Table 3. The dependent variable was government debt risk ($Deb$), with a maximum value of 357.68, a minimum value of 0.48, a mean of 49.01, and a standard deviation of 31.62. These statistics indicated that the variables exhibited considerable variation and representativeness.

For the key explanatory variable, public participation in budgeting ($Part$), the maximum value was 100, the minimum value was 0, the mean was 20.98, and the standard deviation was 19.03, reflecting significant variability.

### 4.2 Regression analysis

**4.2.1 Benchmark regression.** Through the Hausman test, this study finds that the fixed effects regression model provides significantly better results than both the random and mixed effects models. Therefore, we adopted a fixed-effects regression model for our regression analysis. Table 4 presents the benchmark regression results for the effect of budgetary participation opportunities on government debt risk.

We present simple univariate regressions with individual fixed effects but no control variables in Model (1) of Table 4. The core variable exhibits a significant negative coefficient of −0.403 (p < 0.01), indicating that public budget participation is associated with a reduction in government debt risk. To fully specify model (1), we incorporated control variables. The results of this complete specification are reported in Table 4. Specifically, the key independent variable ($Part$), which represents public concern about government debt, is significantly negatively correlated with government debt risk ($Deb$) at the 1% level. Among the control variables, legislative supervision and the economic growth rate are all negatively correlated with government debt risk, suggesting that higher levels of internal supervision within a country are associated with lower government debt risk. The urbanization rate is positively correlated with the local government debt risk at the 1% significance level, indicating that as urban populations and scales expand, government investments in infrastructure, education, and healthcare also increase. Constrained fiscal revenue may motivate the government to use debt to bridge the gap between fiscal revenue and expenditure, thereby increasing the government debt ratio. This finding aligns with the existing

**Table 3. Descriptive statistics of main variables.**

| Variable names | Observations | Mean | Standard deviation | Minimum | Maximum value |
|---|---|---|---|---|---|
| Deb | 664 | 49.01 | 31.62 | 0.48 | 357.68 |
| Part | 664 | 20.98 | 19.03 | 0 | 100 |

**Table 4. The benchmark regression results.**

| Dependent Variable | deb | |
| --- | --- | --- |
| Model | (1) | (1) |
| Part | −0.403*** (0.089) | −0.27*** (0.083) |
| LnGNI | NO | −5.53 (6.57) |
| Leg | NO | − 0.15** (0.059) |
| Aud | NO | −0.07 (0.093) |
| Rgdp | NO | −0.95*** (0.378) |
| Urb | NO | 3.07*** (0.614) |
| Cul | NO | −0.07 (0.227) |
| Observations | 664 | 664 |

Note: ***, **, and * denote p < 0.01, p < 0.05, and p < 0.1, respectively; values in parentheses are cluster-robust standard errors.

research conclusions. Although the educational attainment of residents, audit supervision and per capita national income were not significantly correlated with government debt risk, these variables were retained in the model for completeness. These regression results support Hypothesis 1 but reject Hypothesis 2, indicating that public budget participation is negatively correlated with government debt risk.

### 4.2.2 Robustness analysis.

(1)　Substitution of Dependent Variables Method

To further validate the robustness of our findings, we conducted a robustness test by substituting alternative dependent variables. Specifically, in addition to the government debt ratio, we employ two additional measures: the Debt/revenue Ratio (*Deb_rev*) and Debt/Spending Ratio (*Deb_spe*). These alternative indicators offer a more comprehensive assessment of the government debt risk. The regression results for this substitution are shown in Table 5.

According to the regression results presented in Table 5, substituting alternative dependent variables reveals that public budget participation continues to have a significant inhibitory effect on government debt risk. The regression outcomes remained largely consistent with those reported in Table 4, indicating that the model is robust.

(2)　Addressing Endogeneity Issues

With regard to public governance, there are potential endogeneity issues owing to the bidirectional causal relationship between PB and government debt risk. On one hand, to address the government's dual role as both "athlete and referee" and to strengthen external supervision, PB mechanisms have been introduced to curb government spending deviations and mitigate government debt risks. However, increased government debt risk can harm overall public welfare, which may stimulate greater public participation in budgeting, thereby exerting external pressure to suppress government debt risk. Thus, a bidirectional causal relationship may exist between PB and government debt risk, leading to endogeneity.

To address this issue, we employ the one-period lag of the core explanatory variable *L_part* as an instrumental variable, following Sun et al. [40]. The reasons for using *L_part* as an instrumental variable are as follows:

**Table 5. Robustness Test Using Alternative Dependent Variables.**

| Dependent Variables | Deb_rev | Deb_spe |
|---|---|---|
| Part | −0.021** | −0.015** |
| | (0.009) | (0.005) |
| LnGNI | −2.033** | −1.515** |
| | (0.988) | (0.679) |
| Leg | −0.020** | −0.011*** |
| | (0.007) | (0.004) |
| Aud | −0.017 | −0.010** |
| | (0.012) | (0.007) |
| Rgdp | −0.111** | −0.050* |
| | (0.047) | (0.028) |
| Urb | 0.285*** | 0.214*** |
| | (0.093) | (0.068) |
| Cul | −0.006 | −0.007 |
| | (0.020) | (0.017) |

Budget participation in the previous period not only influences the breadth and depth of "government empowerment" but also provides a learning opportunity for the public. These experiences, in turn, affect the level of future budget participation. Thus, the variable *L_part* is highly correlated with the dependent variable *Deb*. *L_part* is not affected by unobservable factors or omitted variables that could influence budget participation levels, ensuring that the error term $\varepsilon$ is not correlated with *L_part*. This makes *L_part* a suitable instrumental variable.

To address potential endogeneity issues, we employ instrumental variable (IV) regression. The F-statistic from the first-stage regression is 81.33, which exceeds the conventional threshold of 10, thereby confirming that *L_part* is not a weak instrument. This validates our choice of instrumental variables. The IV regression results are presented in Table 6.

Table 6 shows that the instrumental variable regression results are largely consistent with the benchmark regression results presented in Table 4, indicating that the instrumental variable regression results used in this study are robust.

## 5 Mechanism analysis

### 5.1 Mechanism analysis and research hypotheses

**5.1.1 Mechanism analysis.** Although PB has a positive effect on mitigating government debt risk, information asymmetry, as a critical factor permeating the entire debt lifecycle, significantly undermines budgetary participation efficacy while exacerbating risk accumulation [41,42]. Within the principal-agent framework of sovereign debt, information asymmetry manifests as a fundamental disparity in information endowment between the public (principal) and the government (agent). This imbalance systematically undermines the efficacy of PB through both moral hazard and adverse selection mechanisms, thereby accelerating risk accumulation, which manifests as a supply-demand dual structure of information asymmetry:

Supply-side failure: Official government balance sheets inadequately reflect debt complexity, fail to incorporate forward-looking liability risks, and lack systematic performance data.

Demand-side suppression: Prioritizing political stability and electoral gains, governments selectively disclose contingent liabilities (e.g., pension gaps, guarantees, public-private partnership (PPP) risks) with opacity and strategically exclude key risk indicators from debt reporting. These dynamics are illustrated in Fig 1.

Complete and accurate government financial disclosures, including on/off balance sheet liabilities, debt structure granularity, debt serviceability metrics, and contingent obligations, are prerequisites for effective civic monitoring of fiscal

**Table 6. Instrumental variable regression results.**

| Dependent Variable | Deb | Z value |
|---|---|---|
| Instrumental variables: L_part | −0.198** (0.098) | −2.01 |
| LnGNI | −4.779 (6.346) | −0.75 |
| Leg | −0.165*** (0.058) | −2.82 |
| Aud | −0.081 (0.093) | −0.87 |
| Rgdp | −0.961** (0.380) | −2.53 |
| Urb | 3.09*** (0.610) | 5.06 |
| Cul | −0.067 (0.234) | −0.29 |

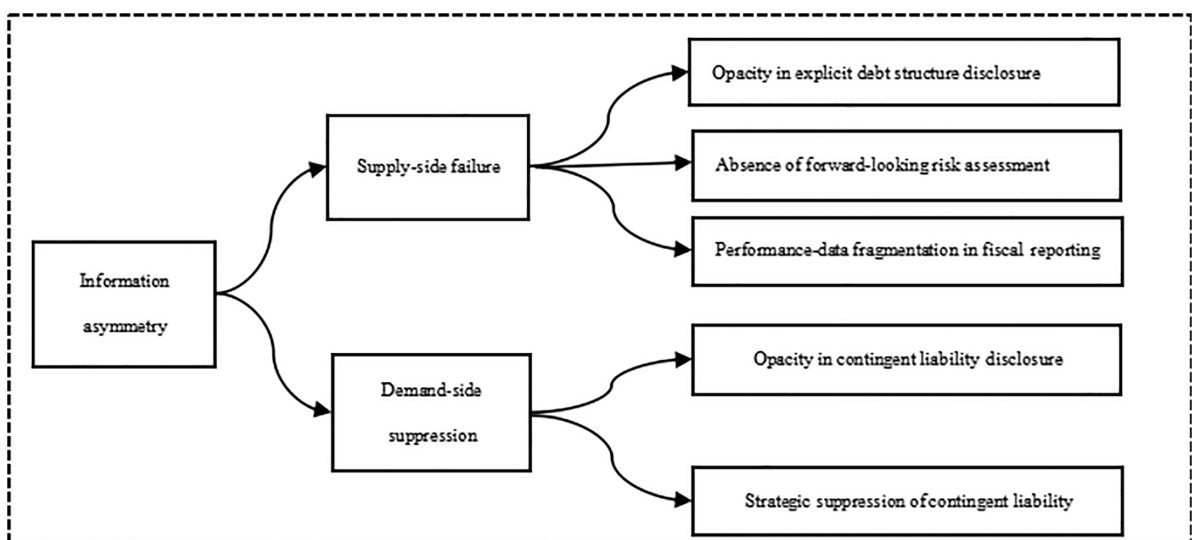

**Fig 1. The Impact of Information Asymmetry on Participatory Budgeting.**

governance [43]. Moreover, when governments ensure timely transparency regarding fiscal risk exposure, this serves as a critical mechanism for reducing the information asymmetries inherent in PB systems [44,45].

Its effectiveness stems from the theoretical logic of "budget participation → improved disclosure transparency (for explicit and contingent liabilities) → reduced information asymmetry → mitigated government debt risk." This mechanism is shown in Fig 2.

**5.1.2 Research hypotheses.** As illustrated in Fig 2, budget participants enhance fiscal transparency through the following budget formulations:

(1) Increasing visibility into the current-year scale, structure, and interest-servicing mechanisms of explicit liabilities.

(2) Identifying aggregate-contingent liabilities and their associated risk levels through budgetary oversight.

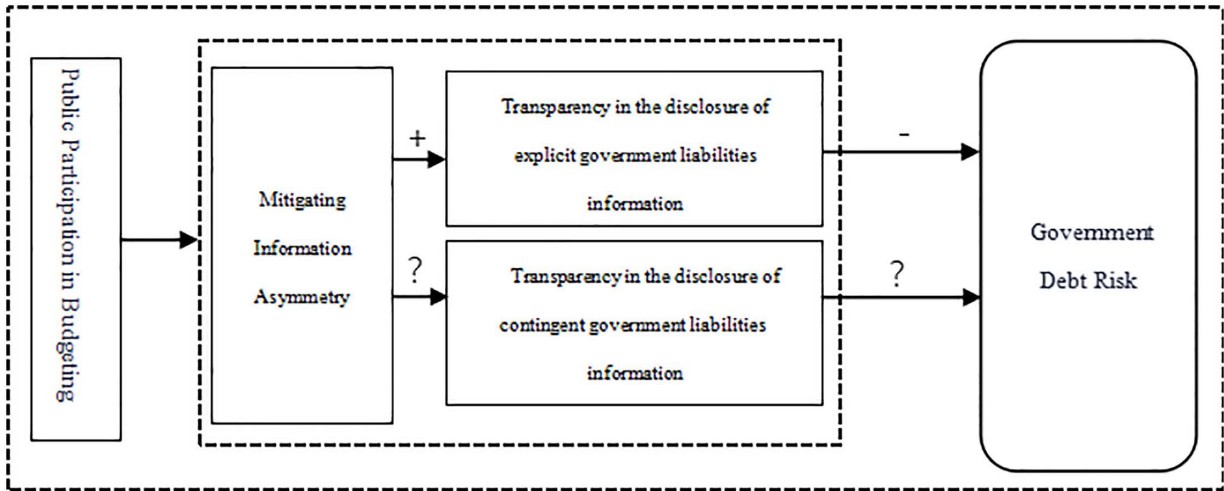

**Fig 2. Theoretical analysis framework of Mitigating Information Asymmetry and Containing Fiscal Risk.**

According to the circumstances triggering repayment, government liabilities fall into two categories: explicit liabilities (covering both explicit and implicit direct liabilities) and contingent liabilities (covering both explicit and implicit contingent liabilities), as defined [46]. The transparency of explicit liabilities, which are directly observable and measurable from a government perspective, operates through at least two key mechanisms. First, signaling theory suggests that high-quality fiscal disclosures function as credible signals of administrative competence and fiscal responsibility [47]. Such transparency enhances policymakers' reputational capital by demonstrating effective governance to voters and credit markets [48]. This is particularly salient given that voters prioritize fiscal transparency, especially debt-related accounting information, when evaluating government performance [49]. Second, empirical evidence demonstrates that enhanced government accounting disclosures yield tangible fiscal advantages. High-quality fiscal reporting is correlated with superior bond ratings and reduced borrowing costs [50]. Rigorous accounting transparency diminishes interest expenses and risk premiums on government debt [51]. Substantiating this, a 14–25 basis-point reduction in debt financing costs for U.S. states adhering to Generally Accepted Accounting Principles (GAAP) versus non-adopters was quantified [52]. This nexus was further reinforced by identifying an inverse relationship between governmental debt levels and the quality of municipal accounting disclosures, thereby confirming that transparency in public financial reporting reduces local government debt financing costs [53]. Transparent fiscal information also facilitates capital market access, thereby mitigating debt sustainability risks [54].

From the debt information stakeholders' perspective, transparent fiscal reporting enables a comprehensive assessment of public officials' stewardship competence. Complete and transparent accounting disclosures are imperative to secure electoral advantages in competitive voting contexts [55]. On one hand, government creditors and investors utilize fiscal disclosures to evaluate sovereign fiscal sustainability, thereby informing decisions on debt purchases and lending terms [56]. On the other hand, budget participants, such as the public and taxpayers, can assess government performance based on their perception and review explicit debt information, including balance sheets and income statements from annual government financial reports. This enables them to analyze, evaluate, and raise concerns, questions, and suggestions regarding significant issues. After gathering public opinion, the relevant government agencies invited experts and department heads to provide their responses and feedback. This approach effectively mitigates the risks of "adverse selection" and "moral hazard" stemming from

information asymmetry, promoting Pareto optimality in the provision of public services [57]. Public engagement in budget participation not only restricts the total amount and structure of government spending but also plays a key role in determining the scale and risk management of government debt. Accordingly, we propose the following hypothesis:

Hypothesis 3: Public participation in budgeting mitigates government debt risk by enhancing the transparency of explicit government liabilities.

The complexity of government contingent debt arises because such information depends not only on the reliability of accounting techniques—such as how government-guaranteed debt and obligations arising from public governance are accounted for—but also on subjective judgments made by decision-makers, including estimates of the likelihood that contingent debt will materialize. The very nature of government contingent liabilities—characterized by informality and uncertainty—means that their accounting recognition and risk measurement require specialized fiscal expertise. Consequently, most budget participants (e.g., grassroots people's congress deputies and the general public) cannot identify such obligations [58]. This finding is corroborated by research: citizens' lack of technical knowledge and data access means PB primarily influences explicit expenditure allocation, rather than enabling effective oversight of contingent liabilities [59,60]. A 2022 survey conducted by Shanghai University of Finance and Economics further supports this conclusion: only 12% of the public could correctly identify implicit government debt constrained by both knowledge gaps and fragmented disclosure practices [61]. Similarly, analysis of European PB cases revealed that, while citizen engagement promotes transparency in long-term fiscal risks, it remains ineffective in improving contingent liability disclosure [62]. Moreover, unlike explicit liabilities, contingent liabilities lack fixed disclosure channels (such as balance sheets and supplementary schedules). Due to pressure from performance evaluations and accountability, government officials tend to conceal contingent debt [63]. These factors impede budget participants' access to government-contingent liability information, thereby undermining their capacity to oversee debt risk.

Based on this, we propose the following hypothesis

Hypothesis 4: Due to budget participants' limited understanding of contingent debts and potential government disclosure restrictions, public budget participation may not effectively mitigate government debt risk by enhancing information transparency.

## 5.2 Mediating variables (*M*)

To ensure the transparency of explicit government debt information, the IBP's OBS includes several questions in its budget monitoring questionnaire (In its 2008 iteration, the survey encompassed key aspects of government debt, including the debt stock, new borrowing, interest rates, loan terms, and principal/interest servicing, as detailed in questions 11, 12, 13, 16, 33, 34, and 72), totaling seven items. In 2010, the same set of questions was used (11, 12, 13, 16, 33, 34, 71), again totaling seven items. In 2012, an additional question (104) was added, bringing the total to eight items. In 2015, the number of questions expanded to include 13, 14, 16, 31, 32, 57, 63, 74, 75, 83, and 90, totaling eleven items. From 2017 to 2023, four additional questions (16, 127, 130, 137) were added to the 2015 set, resulting in a total of fourteen items.

Regarding contingent debt information, the OBS questionnaire includes two specific questions (42 and 43) that address issues such as government commitments, guarantees, and the disclosure of principal and interest for other contingent debts. The scores for explicit debt disclosure (*Exp_deb*) and contingent debt disclosure (*Con_deb*) are calculated by averaging the public ratings of the questions in the annual survey (Data source: https://internationalbudget.org/open-budget-survey/download).

## 5.3 Model setting

Drawing on the research of Sun et al. [64], to test Hypotheses 3 and 4, this study constructs the following recursive model:

$$M_{it} = \beta_0 + \beta_1 part_{it} + \sum \beta_j X_{it} + \mu_i + \varepsilon_{it} \tag{2}$$

$$Deb_{it} = \lambda_0 + \lambda_1 part_{it} + \lambda_2 M_{it} + \sum \lambda_j X_{it} + \mu_i + \varepsilon_{it} \tag{3}$$

In Models (2) and (3), the variable (*M*) represents the transparency scores for government explicit debt disclosure (*Exp_deb*) and contingent debt disclosure (*Con_deb*). The definitions of the remaining variables are identical to those in Model (1).

## 5.4 Mechanism test results

To verify Hypotheses 3 and 4, the following conditions must be satisfied.

(1) Budget participation significantly influences government debt risks.

(2) Budget participation significantly affects the explicit debt information disclosure or contingent debt information disclosure quality.

(3) The effect of budget participation on government debt risk becomes insignificant or weakens upon controlling for the intermediary variables of explicit and contingent debt disclosure.

Using explicit and contingent debt disclosure quality (*Exp_deb, Con_deb*) as mediators, we regress Models (2) and (3); results are reported in Table 7.

As shown in Table 7, when explicit government debt disclosure quality (*Exp_deb*) serves as the mediating variable, budget participation (*Part*) significantly enhances disclosure quality at the 1% level. Models (2) and (3) demonstrate that

**Table 7. Mediation Effect Regression Results.**

| Mediating variable | Exp_deb | | Con_deb | |
|---|---|---|---|---|
| **Model** | **(2)** | **(3)** | **(2)** | **(3)** |
| Dependent variable | Exp_deb | Deb | Con_deb | Deb |
| Part | 0.354*** (0.052) | −0.200*** (0.068) | 0.041 (0.087) | −0.277*** (0.084) |
| Exp_deb | | −0.215** (0.088) | | |
| Con_deb | | | | 0.018 (0.043) |
| LnGNI | −2.40 (4.81) | −6.05 (6.720) | 9.11* (4.908) | −5.703*** (6.607) |
| Leg | 0.17** (0.071) | −0.119** (0.058) | 0.121 ** (0.057) | −0.158*** (0.603) |
| Aud | 0.140** (0.071) | −0.044 (0.089) | 0.051 (0.067) | −0.075 (0.093) |
| Rgdp | 0.271 (0.176) | −0.896** (0.348) | 0.34 1* (0.192) | −0.960** (0.381) |
| Urb | −0.047 (0.542) | 3.060*** (0.637) | −0.721 (0.476) | 3.084*** (0.613) |
| Cul | −0.054 (0.104) | −0.088 (0.220) | −0.027 (0.096) | −0.076 (0.228) |

after introducing this mediator, although budget participation's negative coefficient on debt risk persists, its magnitude decreases. Concurrently, the mediating variable retains a significant inhibitory effect on debt risk at the 5% level. The proportion of the mediating effect, at 28.18%, is significant. These results demonstrate that explicit debt disclosure quality (Exp_deb) plays a significant mediating role in government debt risk, supporting Hypothesis 3.

When contingent liability disclosure quality (Con_deb) serves as the mediating variable in Model (2), budget participation (Part) shows no significant effect on Con_deb. Model (3) regression results further indicate that Part significantly affects government debt risk at the 1% level, while the mediating variable (Con_deb) remains statistically insignificant. This verifies Hypothesis 4.

## 6 Further discussion

Existing research indicates that the impact of fiscal transparency on government debt risk varies, depending on the accounting basis used in government accounting [64]. Given high degree of association relationship between government debt information disclosure and accounting bases (According to the classification by the IMF, a country's accounting basis can be divided into the cash basis, the modified cash basis, the modified accrual basis, and the accrual basis), it is reasonable to hypothesize that the effect of budget participation on government debt risk may also exhibit heterogeneity across different accounting bases.

To explore this hypothesis, we categorized the samples based on government accounting using data from the "Global Survey on Central Government Accounting and Reporting" conducted by PricewaterhouseCoopers in April 2013. This survey provides comprehensive statistical data on the accounting practices of 100 countries. The grouping results are listed in Table 8.

The findings of the grouping results for Regression Model (1) are summarized in Table 9.

As shown in Table 9, budget participation maintains a significant inhibitory effect on government debt risk at the 1% level under an accrual or modified accrual accounting basis. Conversely, this inhibitory effect is not statistically significant under a cash or modified cash. The Chow test also revealed heterogeneity between the two groups.

These findings suggest that an accrual or modified accrual basis of accounting can disclose government debt information more comprehensively, thereby enhancing the effectiveness of budget participation in mitigating government debt risk. Conversely, in countries that use a cash or modified cash basis for government accounting, the inability to confirm government debt information accurately and fully limits the impact of budget participation in reducing government debt risk.

**Table 8. Grouping of government accounting bases in global countries.**

| A modified accrual or accrual basis (57) | A full cash or revised cash basis (26) |
|---|---|
| Britain, Czech, Slovenia, Sweden, Peru, France, Turkey, Georgia, Kazakhstan, India, Tanzania, Russia, Saudi Arabia, Colombia, Democratic Republic of the Congo, Honduras, Philippines, Poland, El Salvador, United States, Argentina, Dominican Republic, Nicaragua, Brazil, Costa Rica, Equatorial Guinea, Guatemala, Mexico, Mongolia, New Zealand, Romania, Sri Lanka, China, Cambodia, Liberia, Vietnam, Egypt, Algeria, Nigeria, Cameroon, Ghana, Morocco, Bangladesh, Thailand, Malaysia, Indonesia, Fiji, Pakistan, Albania, North Macedonia, Croatia, Bulgaria, South Africa, Angola, Zambia | Azerbaijan, Chad, Bolivia, Ecuador, Egypt, Montenegro, Germany, Botswana, Jordan, Kyrgyzstan, Burkina Faso, Lebanon, Liberia, Namibia, Niger, Nepal, Norway, Papua New Guinea, SAO Domingo, Senegal, Serbia, South Korea, Tobago, Ukraine, Venezuela, Yemen |

Data Sources: The global government accounting landscape was derived from the PricewaterhouseCoopers (PwC) Survey on Global Central Government Accounting and Reporting conducted in April 2013. This monitoring report presents statistics on the government accounting basis for 2013, and documents the reform progress of the national government accounting basis over the following five years. Data on government accounting from 2019 to 2023 are sourced from the IMF GFS.

**Table 9. Regression results by group.**

| grouping | The modified accrual or accrual basis | The modified cash or cash basis |
|---|---|---|
| Model | (1) | (1) |
| Part | −0.224*** | −0.383 |
| | (0.055) | (0.254) |
| LnNGI | 0.190 | −13.276 |
| | (6.43) | (13.87) |
| Leg | −0.136** | −0.182 |
| | (0.062) | (0.139) |
| Aud | 0.053 | −0.315** |
| | (0.077) | (0.245) |
| Rgdp | −0.419* | −1.451** |
| | (0.211) | (0.661) |
| Urb | 3.184*** | 1.761 |
| | (0.596) | (1.766) |
| Cul | −0.158 | 0.207 |
| | (0.191) | (0.753) |
| Observations | 456 | 208 |

# 7 Conclusion and policy recommendations

## 7.1 Research conclusions

This study employs international panel data (2008–2023) to investigate the relationship between public budget participation and government debt risk. Key findings are:

(1) The analysis documents that, on average, higher public budget participation is associated with a marked mitigation of government debt risk. It should be noted that this finding captures an average trend, and there is likely heterogeneity in the strength of this relationship across different national contexts.

(2) The quality of explicit debt disclosure mediates the curbing effect of budget participation on debt risk. Conversely, contingent debt disclosure quality shows no significant mediating effect, suggesting the widespread inadequate disclosure of contingent liabilities. Adopting more reliable accounting methods to disclose these liabilities is crucial for enhancing the decision usefulness of government financial reporting.

(3) The debt risk-mitigating effect of budget participation is stronger under accrual-based accounting systems (modified or full accruals) than under cash-based systems (modified cash or cash).

## 7.2 Policy recommendations

Given the effectiveness of budget participation in mitigating government debt-related "gray rhino" risk, this study proposes the following policy recommendations to strengthen its inhibitory effect:

(1) Establish Legal Foundations: Amend the Budget Law to codify "participatory budgeting," defining participant selection, scope of participation, procedures, disclosure requirements, and accountability mechanisms.

(2) Expanding participation scope: Systematically implement participatory budgeting, particularly for livelihood expenditures and major government investments, as a prerequisite for aligning spending priorities and mitigating associated debt risks.

(3) Advanced accounting reform: Modified or full accrual accounting is adopted to enhance the reliability and relevance of financial reporting. Prioritize transparency improvements for government guarantees, commitments, PPPs, and other critical contingent liabilities.

## 8 Limitations and scope for future research

### 8.1 Limitations

**(1) Transparency measurement bias.** Due to the complexity in measuring contingent liabilities and politically motivated concealment of fiscal risks, the current IBP questionnaire fails to accurately capture disclosure transparency. This may cause systematic deviations in empirical results.

**(2) Off-balance-sheet coverage gap.** Prevailing government debt risk metrics primarily rely on explicit liabilities in balance sheets. As widely adopted international standards exclude contingent liabilities from financial statements, risk indicators substantially underestimate exposures.

### 8.2 Scope for future research

**(1) Questionnaire design optimization.** Develop multidimensional scales (e.g., guarantee probability tiers, PPP risk matrices) to construct discriminant-validated transparency assessment tools.

**(2) Accounting standards reform.** Promote on-balance-sheet disclosure of contingent liabilities under accrual-based accounting systems, implementing the IMF (2022) Handbook on Fiscal Risk Analysis Chapter 4 framework for holistic risk surveillance.

**(3) Apply multilevel modeling.** Multilevel modeling offers a powerful way to analyze the hierarchical structure of the data, with subnational entities nested within countries. It is particularly useful for investigating the complex, cross-level relationships between participatory budgeting and subnational government debt risk, as it controls for unobserved heterogeneity at both governmental levels.

## Supporting information

**S1 File. Researcher stataset.**
(XLSX)

## Acknowledgments

We gratefully acknowledge the anonymous reviewers for their valuable insights and Editage (www.editage.com) for professional English language editing.

## Author contributions

**Data curation:** Qianqian Zhang.

**Writing – original draft:** Yongpeng Li.

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
