## [Decision Letter · Decision Letter 0]

25 Jun 2025

Dear Dr. li,

Thank you for submitting your manuscript to PLOS ONE. After careful consideration, we feel that it has merit but does not fully meet PLOS ONE’s publication criteria as it currently stands. Therefore, we invite you to submit a revised version of the manuscript that addresses the points raised during the review process.

We look forward to receiving your revised manuscript.

Kind regards,

Avanti Dey, PhD

Staff Editor

PLOS ONE

Journal Requirements:

3. In the online submission form, you indicated that your data is available only on request from a third party. Please note that your Data Availability Statement is currently missing the contact details for the third party, such as an email address or a link to where data requests can be made. Please update your statement with the missing information.

Reviewers' comments:

Reviewer's Responses to Questions

**Comments to the Author**

1. Is the manuscript technically sound, and do the data support the conclusions?

Reviewer #1: Partly

2. Has the statistical analysis been performed appropriately and rigorously?

Reviewer #1: I Don't Know

3. Have the authors made all data underlying the findings in their manuscript fully available?

Reviewer #1: No

4. Is the manuscript presented in an intelligible fashion and written in standard English?

Reviewer #1: Yes

Reviewer #1: Dear Author,

First of all, I would say that you have an interesting topic of research. In fact, participatory budgeting is an important instrument both for citizens and public management allowing better public budget decisions. So, investigating the factors that affect the public budget and the relevance of participatory budgeting is important. However, at the present stage of the work, I consider that it still requires some improvement. In the following lines, I express my concerns on the paper with the purpose of helping on its development.

My concerns on the paper

+ The abstract does not present the results. It is important to have an abstract that actually comprises the whole paper, mainly the results!

+ The author states “In this process, participants focus not only on the allocation of financial resources and expenditure priorities but also on the sustainability of government debt levels”, citing Park et al. (2023). I confess my doubt if it is indeed a fact/reality that budget participants actually worry on sustainability of government debt levels? If I am not wrong, society members in general seem to worry on their demands and not on government debt levels. Am I wrong? Is there any research about this? The explanation “This is because the scale and risk of government debt impact both the welfare of current budget participants and the tax burden and welfare of future generations” points out that society members would be actually committed to government debt levels. Did the papers referred (Sintomer et al., 2012; Chen et al., 2024) actually ask this to society members [citizens]. Would it be a reality all over the world? The same doubt arises with the statement “Through participatory budgeting, the public can directly review government expenditures and influence the scale and priority of government investment projects, thereby constraining the total amount and structure of government debt”.

+ Until the sentence “In the context of participatory budgeting, information asymmetry remains a critical factor influencing its effectiveness”, the construct “information asymmetry” was not mentioned in the work. Abruptly, it emerges as a core concept influencing the effectiveness of the participatory budgeting. Why is that a reality?

+ Until the model of fig 1, the work makes no reference to the construct “Government Responsiveness”. What does the author mean exactly with “Government Responsiveness”? In my opinion, the explanation of the theoretical framework [fig 1] is not clear.

+ I need to recognize that the rationale presented to support Hypothesis 1 is confusing and not enough. In my view, citizens are focused on their demands. I am not sure about the interest of budget participants actively engaging in negotiating explicit debt stock. Is it a reality? My concern matches the proposal of the Elite Democracy Theory that mentions the challenge of adequate disclosure of government debt information facing the difficult of its comprehension by citizens.

+ Section 3.1.1 seems confusing. The author says that faced limitation on data availability which led him to use Debt-to-revenue ratio. However, right after, he declares that “IMF does not directly provide data on the debt-to-revenue ratio”. So, he computes Debt-to-revenue ratio using debt-to-GDP ratio that was supposed to be unavailable. If debt-to-GDP ratio is a proxy to government debt risk as declared in the paragraph, why isn’t used? The text about the Debt-to-revenue ratio calculus [“The debt-to-revenue ratio is calculated by dividing the outstanding government debt by current government revenue”] does not match with the presented equation: “Debt-to-revenue ratio = (Debt-to-GDP ratio) / (Revenue-to-GDP ratio).” In sum, the work must be more careful about its main variable.

+ In section 3.1.2, why using ordinals as in “114th and 125th”?

+ Most experiences on participatory budget seem to be in city governments. The author declares Budget Participation as being started in Porto Alegre/Brazil. I consider that the proxy for Budget Participation (Parti) needs a more detailed explanation since it seems to be a country assessment. Looking at Brazil, the country mentioned by the author, which comprises 5,570 municipalities, where just a few cities used Budget Participation in the 1990’s and 2000’s, how is such a country, or any other, precisely evaluated by IBP? This is a crucial concept/construct in the work so that its measurement must be very clearly explained.

+ Hypothesis 2 proposes the positive effect of public budget participation on government debt risk taking into account that public budget participation is able to enhance transparency of explicit government debt information. Are citizens so powerful worldwide?

+ If I am not wrong, hypotheses 3a and 3b seem to be contradictory. The author labels them as competitive. Nevertheless, there is no argument about the possible effect of public budget participation improving, or worsening, or not affecting, the transparency of contingent debt information. Competitive hypotheses are better suited when there are contradictory/competitive arguments. About “contingent debt information”, are they disclosed/withheld due to the measurement uncertainties or lack of government interest in disclosing it? In any case, considering the work proposal about an active behavior of citizens on budget participating, it seems to be expected that public engagement would be able to foster the enhancement of transparency of contingent debt information.

+ The author mentions “Mediating Variables” in section 3.1.3. However, in the theoretical background, or even in section 3.1.3, as far as I could see, there is no reference to any mediating effect. When the reader sees it, he guesses that it can be related to Hypotheses 2 and 3 but he is not sure. Only in section 3.2, the reader sees models taking into account the possible effect of Budget Participation (Parti) on government explicit debt disclosure (Expli_deb) and contingent debt disclosure (Conti_deb) that are proposed as mediating variables [M]. Why does the author propose a mediating effect that he has no idea on such effect? The author states “the signs of other coefficients (β1, λ1, λ2) remain to be determined”. According to this statement, the author has no idea about the effect of Budget Participation (Parti) on the mediating variables and of the mediating variables on the main element of observation “government debt risk” (Deb). Considering this situation, I am not sure about the suitability of H2 and H3. In fact, there should be a kit of models for H2 and another for H3. In H2, it seems that the author suggests the positive effect what is different of H3. However, in section 3.2 he does not distinguish them [H2 and H3].

+ Format of table 1 deserves improvement so that variable category may be better identified.

+ When referring to model estimates in tables it would be much better referring to model labels instead of column tables as in “In column 2 of Table 3”.

+ In Section 3.1.1, the author says that faced limitation on data availability, as I commented before. However, in section 4.3.1, the author uses alternate constructs for debt risk. Isn’t contradictory. Does the research actually face limitation on data availability.

+ Results on contingent debt disclosure (Conti_deb) that are proposed as mediating variables [M] are not subject to assessment given the abovementioned problems on it. Any result found is valid.

As a whole, I say that the research topic is interesting. However, in the present stage of the study, I consider that these theoretical/methodological/analysis deficiencies compromise the paper quality. I suggest the author to go deeper in the literature and rethink some crucial elements: theoretical foundations; make an effort to present suitable research hypotheses 2 and 3; improve results presentation; and explain completely the proxy for public budget participation.

I hope that my comments are useful in the development of the work.

Best Regards

**Do you want your identity to be public for this peer review?** For information about this choice, including consent withdrawal, please see our Privacy Policy

Reviewer #1: No

---

## [Author Response · Author response to Decision Letter 1]

13 Sep 2025

1.The abstract does not present the results. It is important to have an abstract that actually comprises the whole paper, mainly the results!

Our response:

We thank the reviewers for their valuable feedback. In response to your comments, we have revised the abstract as follows:

(1) Added key research conclusions, which can be found in lines 22 to 29 of the abstract;

(2) Explicitly stated the objective and method of this study, as detailed in lines 6 to 20.

2.The author states “In this process, participants focus not only on the allocation of financial resources and expenditure priorities but also on the sustainability of government debt levels”, citing Park et al. (2023). I confess my doubt if it is indeed a fact/reality that budget participants actually worry on sustainability of government debt levels? If I am not wrong, society members in general seem to worry on their demands and not on government debt levels. Am I wrong? Is there any research about this? The explanation “This is because the scale and risk of government debt impact both the welfare of current budget participants and the tax burden and welfare of future generations” points out that society members would be actually committed to government debt levels. Did the papers referred (Sintomer et al., 2012; Chen et al., 2024) actually ask this to society members [citizens]. Would it be a reality all over the world? The same doubt arises with the statement “Through participatory budgeting, the public can directly review government expenditures and influence the scale and priority of government investment projects, thereby constraining the total amount and structure of government debt”.

Our response:

We sincerely appreciate the reviewer’s insightful comments and fully agree with the points raised. As rightly noted, the behavior of budget participants is often closely linked to their personal interests. In response, we have expanded our discussion in the revised manuscript to thoroughly examine how government debt arrangements affect participants’ expected income, employment prospects, and sense of benefit. We have also systematically examined arguments from both supporting and opposing perspectives, leading to the formulation of a set of competitive hypotheses. The specific revisions can be found in Lines 67–135 of the revised manuscript.

3.Until the sentence “In the context of participatory budgeting, information asymmetry remains a critical factor influencing its effectiveness”, the construct “information asymmetry” was not mentioned in the work. Abruptly, it emerges as a core concept influencing the effectiveness of the participatory budgeting. Why is that a reality?

Our response:

We thank the reviewers for their comment. We agree that the concept of information asymmetry was initially introduced without sufficient context. In response, we have added a theoretical foundation regarding its role in budget participation and have developed Figure 1 to illustrate the mechanism. The specific revisions can be found in lines 332 to 348 of the revised manuscript.

Figure 1 is provided below:

4.Until the model of fig 1, the work makes no reference to the construct “Government Responsiveness”. What does the author mean exactly with “Government Responsiveness”? In my opinion, the explanation of the theoretical framework [fig 1] is not clear.

Our response:

We thank the reviewer for this valuable feedback. We fully agree that the original introduction of the concept of "government response" appeared somewhat abrupt, largely due to the difficulty in quantitatively capturing the government’s responsive behavior to budget participants’ concerns.

Instead, we argue that enhancing the transparency of debt information—including both explicit and contingent liabilities—enables the government to substantively address public concerns and mitigate information asymmetry.

In alignment with the theoretical framework illustrated in Figure 1, we have revised the original concept of "government response" to "reduction of information asymmetry" and updated the figure accordingly. This revision provides clearer theoretical support for the pathway:

budget participation → improved disclosure transparency (of explicit and contingent liabilities) → reduced information asymmetry → mitigated government debt risk.

The updated Figure 2 is presented below:

5.I need to recognize that the rationale presented to support Hypothesis 1 is confusing and not enough. In my view, citizens are focused on their demands. I am not sure about the interest of budget participants actively engaging in negotiating explicit debt stock. Is it a reality? My concern matches the proposal of the Elite Democracy Theory that mentions the challenge of adequate disclosure of government debt information facing the difficult of its comprehension by citizens.

Our response:

We appreciate the reviewer’s comment and fully agree with their perspective. As also addressed in our response to Comment #2, we have incorporated an analysis based on the concept of budget participants as rational economic agents. The revision examines how government debt arrangements may affect participants' income, employment, and perceived sense of benefit. Detailed modifications can be found in lines 67 to 135 of the revised manuscript.

6.Section 3.1.1 seems confusing. The author says that faced limitation on data availability which led him to use Debt-to-revenue ratio. However, right after, he declares that “IMF does not directly provide data on the debt-to-revenue ratio”. So, he computes Debt-to-revenue ratio using debt-to-GDP ratio that was supposed to be unavailable. If debt-to-GDP ratio is a proxy to government debt risk as declared in the paragraph, why isn’t used? The text about the Debt-to-revenue ratio calculus [“The debt-to-revenue ratio is calculated by dividing the outstanding government debt by current government revenue”] does not match with the presented equation: “Debt-to-revenue ratio = (Debt-to-GDP ratio) / (Revenue-to-GDP ratio).” In sum, the work must be more careful about its main variable.

Our response:

We thank the reviewer for their constructive feedback. Internationally recognized metrics for assessing government debt risk include the "debt-to-GDP ratio" and the "debt-to-revenue ratio." In accordance with the reviewer’s suggestion, we have adopted the "debt-to-GDP ratio" as the core proxy variable for government debt risk across all regression analyses, while also incorporating the "debt-to-revenue ratio" and the "debt-to-spending ratio" as alternative measures for supplementary robustness checks. The corresponding revisions are presented in Tables 3 to 5 of the revised manuscript.

7.In section 3.1.2, why using ordinals as in “114th and 125th”?

Our response:

We thank the reviewer for the valuable comments. We have revisited Section 3.1.2 and confirmed that the original text stated:

“For the 2006–2012 period, the survey included questions numbered 114 to 125 pertaining to the opportunity score, amounting to 12 questions in total.”

The Open Budget Survey by IBP comprises 142 questions in overall, among which Questions 114 to 125 specifically assess public participation in the budget process. The remaining questions focus on issues related to government fiscal transparency.

8.Most experiences on participatory budget seem to be in city governments. The author declares Budget Participation as being started in Porto Alegre/Brazil. I consider that the proxy for Budget Participation (Parti) needs a more detailed explanation since it seems to be a country assessment. Looking at Brazil, the country mentioned by the author, which comprises 5,570 municipalities, where just a few cities used Budget Participation in the 1990’s and 2000’s, how is such a country, or any other, precisely evaluated by IBP? This is a crucial concept/construct in the work so that its measurement must be very clearly explained.

Our response:

We fully understand and appreciate the reviewer’s concerns. Indeed, the Open Budget Survey conducted by the IBP is primarily carried out at the national level, and its evaluation scores reflect the overall budget transparency of countries. Taking the 2023 survey as an example, it included a total of 18 questions related to public participation in the budgeting process, numbered from 125 to 142, which specifically cover:

1�Does the executive use participation mechanisms through which the public can provide input during the formulation of the annual budget (prior to the budget being tabled in parliament)?

2�With regard to the mechanism identified in question 125, does the executive take concrete steps to include vulnerable and under-represented parts of the population in the formulation of the annual budget?

3�"During the budget formulation stage, which of the following key topics does the executive’s engagement with citizens cover?�including�1. Macroeconomic issues� 2. Revenue forecasts, policies, and administration� 3. Social spending policies� 4. Deficit and debt levels� 5. Public investment projects� 6. Public services

4�Does the executive use participation mechanisms through which the public can provide input in monitoring the implementation of the annual budget?

5�With regard to the mechanism identified in question 128, does the executive take concrete steps to receive input from vulnerable and underrepresented parts of the population on the implementation of the annual budget?

6�"During the implementation of the annual budget, which of the following topics does the executive’s engagement with citizens cover?

7�"When the executive engages with the public, does it provide comprehensive prior information on the process of the engagement, so that the public can participate in an informed manner?

8�With regard to the mechanism identified in question 125, does the executive provide the public with feedback on how citizens’ inputs have been used in the formulation of the annual budget?

9�With regard to the mechanism identified in question 128, does the executive provide the public with information on how citizens’ inputs have been used to assist in monitoring the implementation of the annual budget?

10�Are participation mechanisms incorporated into the timetable for formulating the Executive’s Budget Proposal?

11�Do one or more line ministries use participation mechanisms through which the public can provide input during the formulation or implementation of the annual budget?

12�Does the legislature or the relevant legislative committee(s) hold public hearings and/or use other participation mechanisms through which the public can provide input during its public deliberations on the formulation of the annual budget (pre-budget and/or approval stages)?

13�"During the legislative deliberations on the annual budget (pre-budget or approval stages), which of the following key topics does the legislature’s (or relevant legislative budget committee) engagement with citizens cover?

14�Does the legislature provide feedback to the public on how citizens’ inputs have been used during legislative deliberations on the annual budget?

15�Does the legislature hold public hearings and/or use other participation mechanisms through which the public can provide input during its public deliberations on the Audit Report?

16�Does the Supreme Audit Institution (SAI) maintain formal mechanisms through which the public can suggest issues/topics to include in the SAI’s audit program (for example, by bringing ideas on agencies, programs, or projects that could be audited)?

17�Does the Supreme Audit Institution (SAI) provide the public with feedback on how citizens’ inputs have been used to determine its audit program?

18�Does the Supreme Audit Institution (SAI) maintain formal mechanisms through which the public can contribute to audit investigations (as respondents, witnesses, etc.)?”

Each of the above questions offers four response options. The Open Budget Survey employs a randomized survey methodology to score citizens’ responses, which are then validated through an independent expert review to ensure objectivity. Supporting documentation for the scoring is also maintained. These measures collectively enhance the accuracy and reliability of the opportunity score for public participation in budgeting.

Additionally, detailed explanations regarding the variables related to participatory budgeting have been added and revised in Lines 173–194 of the updated manuscript. Please refer to this section for further information.

9.If I am not wrong, hypotheses 3a and 3b seem to be contradictory. The author labels them as competitive. Nevertheless, there is no argument about the possible effect of public budget participation improving, or worsening, or not affecting, the transparency of contingent debt information. Competitive hypotheses are better suited when there are contradictory/competitive arguments. About “contingent debt information”, are they disclosed/withheld due to the measurement uncertainties or lack of government interest in disclosing it? In any case, considering the work proposal about an active behavior of citizens on budget participating, it seems to be expected that public engagement would be able to foster the enhancement of transparency of contingent debt information.

Our response:

We sincerely thank the reviewers for their insightful comments. We fully agree with your observation regarding the limitations of the original competing hypothesis. In response, we have implemented the following revision:

The competing hypothesis has been reformulated and relabeled as Hypothesis 4, which now states:

“Hypothesis 4: Owing to the measurement uncertainties inherent in contingent debt information, public budget participation may not effectively mitigate government debt risk through enhanced transparency of such information.”

This new hypothesis is accompanied by a thorough discussion and justification. The corresponding revisions can be found in lines 412 to 439 of the revised manuscript.

10.Hypothesis 2 proposes the positive effect of public budget participation on government debt risk taking into account that public budget participation is able to enhance transparency of explicit government debt information. Are citizens so powerful worldwide?

Our response:

We greatly appreciate the reviewers’ valuable comments. As an important supplement to elite democracy, participatory budgeting is increasingly being adopted and developed worldwide. Although current practices of participatory budgeting face challenges such as insufficient participation and elite capture, substantial evidence has demonstrated that this mechanism can effectively promote the engagement of ordinary citizens in national budget decision-making, thereby representing a significant innovation in governance models. For instance, in China, cities such as Wenling in Zhejiang Province, Jiaozuo in Henan Province, and Shanghai have implemented participatory budgeting and achieved positive outcomes. Democratic processes are an indispensable part of national governance. Despite existing difficulties and challenges, we believe that the inherent vitality and demonstrative effect of participatory budgeting will gradually become evident, ultimately contributing to the broader practice and development of democratic governance.

11.The author mentions “Mediating Variables” in section 3.1.3. However, in the theoretical background, or even in section 3.1.3, as far as I could see, there is no reference to any mediating effect. When the reader sees it, he guesses that it can be related to Hypotheses 2 and 3 but he is not sure. Only in section 3.2, the reader sees models taking into account the possible effect of Budget Participation (Parti) on government explicit debt disclosure (Expli_deb) and contingent debt disclosure (Conti_deb) that are proposed as mediating variables [M]. Why does the author propose a mediating effect that he has no idea on such effect? The author states “the signs of other coefficients (β1, λ1, λ2) remain to be determined”. According to this statement, the author has no idea about the effect of Budget Participation (Parti) on the mediating variables and of the mediating variables on the main element of observation “government debt risk

---

## [Decision Letter · Decision Letter 1]

14 Nov 2025

Dear Dr. li,

Thank you for submitting your manuscript to PLOS ONE. After careful consideration, we feel that it has merit but does not fully meet PLOS ONE’s publication criteria as it currently stands. Therefore, we invite you to submit a revised version of the manuscript that addresses the points raised during the review process.

We look forward to receiving your revised manuscript.

Kind regards,

Matteo Fragetta

Academic Editor

PLOS ONE

Journal Requirements:

Reviewers' comments:

Reviewer's Responses to Questions

**Comments to the Author**

Reviewer #2: All comments have been addressed

2. Is the manuscript technically sound, and do the data support the conclusions?

Reviewer #2: Yes

3. Has the statistical analysis been performed appropriately and rigorously?

Reviewer #2: Yes

4. Have the authors made all data underlying the findings in their manuscript fully available?

Reviewer #2: Yes

5. Is the manuscript presented in an intelligible fashion and written in standard English?

Reviewer #2: (No Response)

Reviewer #2: The manuscript deals with an important topic of participatory budgeting (PB) and aims to explore how citizen involvement in budgetary decision-making can mitigate government debt risk across countries. It also explores the mechanisms through which PB affects fiscal risk, focusing on information transparency (explicit and contingent liabilities) as mediating factors. The manuscript is well written, I suggest only minor revisions before it can be accepted for publication.

The authors could clarify the aim by pointing out to the realistic scope of PB’s influence — distinguishing between local participatory processes and national fiscal management systems.

The Introduction section could engage with more literature, e.g. Džinić, J., et al. 2016. Participatory budgeting: a comparative study of Croatia, Poland and Slovakia. NISPAcee Journal of Public Administration and Public Policy, vol. IX, no. 1, p. 31 - 56.; Murray Svidroňová, M., Benzoni Baláž, M., Klimovský, D. and Kaščáková, A. (2024), "Determinants of sustainability of participatory budgeting: Slovak perspective", Journal of Public Budgeting, Accounting & Financial Management, Vol. 36 No. 1, pp. 60-80.; Murray Svidroňová, M., Nikolov, M., Garvanlieva Andonova, V., & Kaščáková, A. (2023). COVID-19 and participatory budgeting in North Macedonia and Slovakia. Public Sector Economics, 47(3), 387-406.

Also, clarify the conceptual boundaries of “participatory budgeting” at national level.

The econometric specifications are standard and competently applied, though results should be interpreted as correlational with causal tendencies, not definitive causal proof.

For future research consider multilevel modelling if future data allow (subnational PB vs. national debt outcomes).

**Do you want your identity to be public for this peer review?** For information about this choice, including consent withdrawal, please see our Privacy Policy

Reviewer #2: No

---

## [Editor Report · Decision Letter 2]

25 Nov 2025

Can Participatory Budgeting Mitigate Government Debt Risk�——An Empirical Analysis Using Cross-national Panel Data

PONE-D-25-21902R2

Dear Dr. li,

We’re pleased to inform you that your manuscript has been judged scientifically suitable for publication and will be formally accepted for publication once it meets all outstanding technical requirements.

Kind regards,

Matteo Fragetta

Academic Editor

PLOS ONE
---

## [Editor Report · Acceptance letter]

PONE-D-25-21902R2

PLOS ONE

Dear Dr. Li,

I'm pleased to inform you that your manuscript has been deemed suitable for publication in PLOS ONE. Congratulations! Your manuscript is now being handed over to our production team.

Kind regards,

on behalf of

Dr. Matteo Fragetta

Academic Editor

PLOS ONE